# Eco-Friendly Epoxy-Terminated Polyurethane-Modified Epoxy Resin with Efficient Enhancement in Toughness

**DOI:** 10.3390/polym15132803

**Published:** 2023-06-24

**Authors:** Kun Zhang, Jinrui Huang, Yigang Wang, Wenbin Li, Xiaoan Nie

**Affiliations:** Key Laboratory of Biomass Energy and Material, Jiangsu Province, Co-Innovation Center of Efficient Processing and Utilization of Forest Resources, Key Laboratory of Chemical Engineering of Forest Products, National Forestry and Grassland Administration, National Engineering Research Center for Low-Carbon Processing and Utilization of Forest Biomass, Institute of Chemical Industry of Forest Products, Chinese Academy of Forestry, Nanjing 210042, China

**Keywords:** epoxy resin, polyurethane, toughness, mechanical properties

## Abstract

Polyurethane is widely used to toughen epoxy resins due to its excellent comprehensive properties and compatibility. However, some demerits of polyurethanes limit their applications, such as the harsh storage condition of isocyanate-terminated polyurethane (ITPU), the limited amount of ITPU in epoxy resin, and using solvents during the preparation of polyurethane-modified epoxy resins. To address these issues, in this study, we reported a facile and green approach for preparing epoxy-terminated polyurethane (EPU)-modified epoxy resins with different EPU contents. It was found that the toughness of the epoxy resin was significantly improved after the addition of EPU. When the EPU content was 30 wt%, the elongation at break and toughness were improved by 358.36% and 73.56%, respectively. In comparison, the toughening effect of EPU outperformed that of ITPU. Moreover, the high content of EPU did not significantly decrease the glass transition temperature and had little effect on the thermal stability of the epoxy resin.

## 1. Introduction

Epoxy resin (EP) has been extensively utilized in various fields, such as machinery manufacturing, aerospace, electronics, and civil construction, due to its exceptional mechanical properties, high adhesion strength, corrosion resistance, electrical insulation, and favorable processing ability [1,2,3,4]. The diglycidyl ether of bisphenol A (DGEBA) is the most widely used epoxy resin due to its low cost and the wide availability of its raw materials, accounting for about 70% of the entire epoxy resin usage [5,6,7]. However, due to its unique chemical structure, the cured DGEBA is brittle and susceptible to cracking under impact, restricting its applications in high-end fields [8,9]. Therefore, there is a pressing need to improve the toughness of DGEBA.

Generally, the toughness of epoxy resin can be improved by adding toughening agents, such as thermoplastic materials [10], hyperbranched polymers [11], core–shell particles [12], liquid rubbers [13], nanofillers [14], and other polymers forming interpenetrating networks (IPNs) or semi-interpenetrating networks (SIPNs) [15,16]. IPNs refer to a class of polymeric system formed by the physical interpenetration of a polymer with no covalent bonds between two or more polymeric networks [17]. Researchers have found that constructing IPN or SIPN structures in the epoxy resin is an efficient method to enhance the toughness of epoxy resins [18,19,20,21].

Polyurethane (PU) is a common polymer for constructing IPN or SIPN structures in epoxy matrix. In 1974, Frisch first reported the modification of EP with PU to form an IPN structure [22]. Since then, PU-modified EP has attracted much attention [23,24,25,26,27]. However, some drawbacks of PU limit its application in the modification of epoxy resin: (1) The storage condition of ITPU is harsh because of the high reactivity of isocyanate. (2) The amount of ITPU for modifying EP is limited due to the high viscosity and easy self-curing at a high content of ITPU (Appendix A). (3) Organic solvents are often required during polyurethane synthesis and the preparation of the modified epoxy resins, which are harmful to humans and can pollution the water, air, soil, etc. [28,29,30,31,32]. Chen et al. synthesized epoxy-terminated polyurethane (EPU) and studied its own properties [33]. Li et al. studied the effect of EPU with various soft segment structures on the toughness of epoxy resin. The results showed that hydroxyl-terminated polyether, as a soft segment of polyurethane, can greatly increase epoxy resin’s toughness [34]. However, no research has been conducted regarding the investigation of the difference between EPU−modified epoxy resins and typical isocyanate-terminated polyurethane−modified epoxy resins to show that the terminal epoxidation of polyurethane is an efficient method for solving the problems of conventional polyurethane-modified epoxy resins.

In this study, an EPU was synthesized by the solventless method, and no solvent was used during the preparation of EPU-modified epoxy resin (Figure 1), which is environmentally friendly. It was found that the addition of EPU was not limited. In this study, the EPU content could reach its highest content (50 wt%). In comparison, liquid epoxy resin without hardener quickly transferred into a solid after the addition of 50 wt% ITPU (Appendix A). The mechanical and thermal properties of the modified epoxy resins with different EPU contents were studied systematically and compared to the epoxy resins modified by ITPU with a similar molecular structure to EPU. It was found that the toughness of the epoxy resin was significantly improved after the addition of EPU and the toughening effect of EPU outperformed that of ITPU. Although the thermal properties of epoxy resins decreased after the addition of the EPU, they were comparable with ITPU-modified epoxy resins at the same modifier content.

## 2. Materials and Methods

### 2.1. Materials

Diglycidyl ether of bisphenol A (DGEBA, epoxide value: ~0.51 mol (100 g^−1^) was supplied by Jiangsu Sanmu Group Co., Ltd. (Yixing, China). Hydroxyl-terminated polyether (DL−220 *M*_w_: ~2000 g mol^−1^) was purchased from Guangzhou Baining New Materials Co., Ltd. (Guangzhou, China). Glycidol (purity 96%) was obtained from Shanghai Aladdin Bio−Chem Technology Co., Ltd. (Shanghai, China). Toluene diisocyanate (TDI) (purity 99%), Dibutyl tin dilaurate (DBTDL) (purity 95%), and polyetheramine (D230, average *M*_n_: ~230 g mol^−1^) were purchased from Shanghai Macklin Biochemical Co., Ltd. (Shanghai, China). Hydrochloric acid (HCl) (purity 37%) and acetone (purity 99%) were purchased from Nanjing Chemical Reagent Co., Ltd. (Nanjing, China). Sodium hydroxide (NaOH) (purity 96%) was received from Xilong Scientific Co., Ltd. (Shantou, China). The chemical structures of DGEBA and polyetheramine D230 are shown in the Appendix A.

### 2.2. Synthesis of Epoxy-Terminated Polyurethane (EPU)

Hydroxyl-terminated polyether DL−220 was dried in a vacuum oven (DZF−6020, Jinghong, China) at 120 °C for 2 h. The dried DL−220 (300 g) and TDI (52.20 g) were heated to 80 °C. After 1 h of reaction, two drops of catalyst DBTDL were added and the reaction was continued until the NCO content decreased to half of the initial (about 2 h) to obtain ITPU. The NCO content was detected in the reaction by using the di N-butylamine technique [35]. Then, 150 g of ITPU and 9.45 g of glycidol were charged into a 250 mL flask with mechanical stirring and the temperature was raised to 80 °C for reaction until the NCO absorption peak (2271 cm^−1^) disappeared in the FT−IR spectra (Figure 2). A sample of the reacting ITPU/glycidol mixture was taken at intervals of 30 min for FT−IR analysis. When the NCO absorption peak disappeared (about 3 h), the reaction was stopped and cooled down to room temperature to obtain EPU.

### 2.3. Preparation of Epoxy Resin and the Modified Epoxy Resin Samples

Firstly, DGEBA/EPU mixtures with different EPU contents were prepared by mechanically mixing DGEBA with EPU via IKA RW20 (IKA, Staufen, Germany) at around 1000 rpm for 10 min. Due to the fact that EPU was used to modify epoxy resin, the highest EPU content was 50 wt%. When the EPU content exceeded 50 wt%, DGEBA was employed to modify the EPU instead, which was contrary to the original goals. Consequently, we set 50 wt% as the maximum EPU content. Based on the EPU content, these samples were named EPU10, EPU20, EPU30, and EPU50. Secondly, according to the epoxy value of different DGEBA/EPU mixtures (Table 1), a predetermined amount of curing agent D230 was added and mixed at ~1000 rpm for 10 min. Finally, the DGEBA/EPU/D230 mixtures were degassed under the vacuum oven (DZF−6020, Jinghong, China) at room temperature. The degassed mixture was poured into the stainless molds and cured at 80 °C for 3 h in a DHG-914CA oven (Jinghong, China). For comparison, a neat epoxy resin without any modifier and a sample with 10 wt% ITPU (ITPU10) were prepared in the same way as the DGEBA/EPU/D230 epoxy resins were.

### 2.4. Characterization

Fourier-transform infrared (FT−IR) spectroscopy was performed using the Nicolet iS50 (Thermo Fisher Scientific, Waltham, MA, USA), with a range of 4000−500 cm^−1^. FT-IR analysis was conducted with a scanning of 16 and a resolution of 4 cm^−1^. The dynamic mechanical analysis (DMA) was performed using the experimental instrument Q800 (TA Instruments, New Castle, DE, USA) in dual cantilever mode from −50 to 150 °C at a heating rate of 3 °C min^−1^ and an oscillating frequency of 1 Hz. The size of samples for the DMA test was 60 mm × 10 mm × 4 mm. The thermostability of materials was analyzed by a thermogravimetric analyzer (TGA), which was performed on a TG209F1 TGA (Ntzsch, Selb, Germany). The TGA test was operated in a nitrogen atmosphere from room temperature to 800 °C at a heating rate of 20 °C min^−1^. The surface morphology of the samples was investigated by a scanning electron microscope (SEM, 3400−I Hitachi, Tokyo, Japan). Surfaces prepared by tensile test fracture samples were used to study the effect of toughening agents on the phase morphology. In order to obtain clearer phase morphology information of the fracture surface, all fracture surfaces were gold-coated with an E-1010 ion sputter (Hitachi, Japan), and the thickness of the coating was 10−20 nm before examination.

The tensile test was carried out on the microcomputer-controlled electronic universal testing machine (LD24.304, Lishi, China) at a crosshead speed of 5 mm min^−1^. According to the international ISO 527−3:2018, dumbbell-shaped samples were prepared. From the viewpoint of energy, the integral area of the stress−strain curve can be used to represent the fracture absorption energy of the tensile test [23]. The toughness of each sample was obtained by integrating the area under the stress−strain curves. The Young’s modulus was calculated from the slope between the stress and the strain, which can be directly obtained from the software of the microcomputer−controlled electronic universal testing machine. Three-point bending experiments were carried out at a stress loading speed of 5 mm min^−1^ according to ISO 178:2019. Three−point bending specimens (100 mm × 10 mm × 4 mm) were used to determine flexural strength. The equipment for the bending test was the same as that for the tensile test. All mechanical tests were performed at 25 °C. At least five replicates were tested for each group of samples and average values were reported.

## 3. Results

### 3.1. Synthesis and Structural Characterization of EPU

The structures of TDI, DL−220, and EPU were analyzed by FT−IR. Figure 2 presents the FT-IR spectra of DL−220, TDI, ITPU, and EPU. Firstly, the terminal hydroxyl group in DL−220 reacts with TDI to obtain ITPU, so the hydroxyl absorption peak at 3486 cm^−1^ in DL−220 disappears after the reaction and a new carbamate group (3300 cm^−1^, 1530 cm^−1^, and 1720 cm^−1^) appears. Secondly, the isocyanate group at the end of ITPU reacts with glycidol to obtain EPU. Therefore, after the reaction, the isocyanate absorption peak of ITPU at 2271 cm^−1^ disappears and an epoxy group (910 cm^−1^) appears. The results of structural characterization prove the successful synthesis of EPU.

### 3.2. Dynamic Mechanical Analysis (DMA)

The viscoelastic properties of the modified epoxy resins were determined by DMA (Figure 3). Similar to the neat epoxy samples, modified epoxy samples with a low amount of EPU exhibit only one step and one peak on the storage modulus and the tan delta versus temperature curves, respectively. However, when the mass ratio of the modifier and DGEBA is 1:1, a second step and second peak appear at a lower temperature, which is associated with the movement of polyether flexible chains at lower temperatures. Tan delta is the ratio of the loss modulus to the storage modulus, and the temperature of its maximum value is taken as the glass transition temperature (*T*_g_). From the tan delta versus temperature curves (Figure 3b), it is found that *T*_g_ decreases gradually as the EPU content increases. This trend can be interpreted by the variation in cross-linking density (*ν*), which is calculated from the rubber elasticity equation as follows [36]:(1)$$\nu =\frac{E\prime }{3RT}$$
where *R* is the gas constant, and *T* and *E*′ are the absolute temperature and storage modulus at *T*_g_ + 30 °C, respectively. The cross-linking densities for the neat epoxy resin and the modified epoxy resin are shown in Table 2. Generally, a higher ν implies more cross−linked bonds in a unit volume, which can restrict the movement of chains, thus resulting in a higher *T*_g_. Consistent with the variation trend of *ν*, *T*_g_ declines with increasing EPU content.

The storage modulus refers to the energy stored due to elastic deformation when the material is deformed, which can be used to reflect the elasticity of the material and characterize the stiffness of the material. As shown in Figure 3a, in the glass state, the storage modulus of neat epoxy resin changes little with temperature. But the storage modulus of the modified epoxy resin decreases with the increase in temperature in the glassy state. And the change trend becomes obvious with the increase in EPU content. In particular, the storage modulus of EPU30 is 1340 MPa at 25 °C, which is 56.8% of that of the neat epoxy. This shows that the addition of EPU can significantly reduce the stiffness of the material.

Although the introduction of EPU into DGEBA does not cause phase separation (Figure 3b), it still affects the original network, especially the distance between the crosslinking points. The half peak width (HPW) of the peak associated with *T*_g_ in the tan delta curve can reflect the dispersion of chain segment motion. As shown in Table 2, adding EPU to the epoxy resin can increase the HPW value, indicating a greater dispersion of the modified sample’s segment length. This is because the original neat epoxy curing network only has DGEBA and D230 molecules, resulting in only one arrangement. However, after the introduction of EPU, there are multiple arrangements. It results in a greater dispersion of chain segment length, thus increasing the HPW value of the modified samples. Furthermore, the decrease in crosslinking density makes the chain segment relaxation easier for the modified samples, resulting in a lower value of peak tan delta than that of neat epoxy resin.

### 3.3. Mechanical Properties

Mechanical properties of the EP/EPU samples were characterized by tensile tests and three-point bending tests (Figure 4), and the data are listed in the Appendix A. As illustrated in Figure 4a,b, elongation at break and toughness are enhanced greatly with increasing EPU content. Notably, the elongation at break of the modified epoxy resin with 10 wt% EPU is 11.02 ± 0.37%, which is a 67.48% improvement compared to that of pure epoxy resin (6.58 ± 0.23%). The toughness of 10 wt% EPU-modified epoxy resin is 4.22 ± 0.36 MJ m^−3^, which is 28.27% higher than that of the neat epoxy resin (3.29 ± 0.24 MJ m^−3^). To highlight the effect of EPU on toughness, ITPU10 was set as the control group, which contains 10 wt% ITPU. ITPU is linked to the internal epoxy network by pre-reaction with the hydroxyl group of DGEBA (Figure 5a), which will retard the decrease in the crosslinking density (Table 2). As a result, it is found that ITPU (elongation at break: 5.54 ± 0.42%, toughness: 3.05 ± 0.20 MJ m^−3^) does not have a toughening effect at such a low dosage. Comparing the same amount of EPU and ITPU, the elongation at break of ITPU-modified epoxy resin is only half of that of EPU-modified epoxy resin and the toughness of ITPU-modified epoxy resin is much lower than that of EPU-modified epoxy resin. Unlike the change trend of elongation at break and toughness, other properties such as tensile strength (Figure 4c), Young’s modulus (Figure 4d), and flexural strength (Figure 4e) of 10 wt% ITPU-modified epoxy resin show an upward trend or a slow decrease trend. The tensile strength is raised by 10.5% as compared to the neat epoxy. The Young’s modulus and flexural strength are reduced by 6.77% and 7.04%, respectively. However, EPU shows a quick decline trend. When the EPU content is 10 wt%, the tensile strength, Young’s modulus, and flexural strength decrease by 20.14%, 16.36%, and 17.98%, respectively.

The difference between EPU and ITPU is mainly due to the different connection modes of EPU and ITPU with epoxy resin (Figure 5). ITPU is directly connected to the DGEBA (Figure 5a). This internal connection will hold the crosslinking density (Table 2), which will limit the movement of the flexible polyether chain. Therefore, this internal connection helps to maintain high tensile strength, Young’s modulus, and flexural strength, but not to improve the ductility and toughness significantly. EPU is externally connected to the epoxy resin curing network through connection with the curing agent (Figure 5b). This connection will reduce the crosslinking density of the material (Table 2), resulting in fewer chemical crosslinking spots per unit volume, which means that connections between segments are weakening. As a result, when the material matrix is subjected to external forces, there is less interaction between the segments, making it easier for the segments to be oriented and move in the direction of the force. Therefore, when the material is subjected to external forces, the initially cured EPU long chain extends in the direction of the force, which is macroscopically manifested by the necking phenomenon of the material. The process contributes to the increase in elongation at break and toughness, thus improving the ductility and toughness of the epoxy resin (Figure 4a,b).

As depicted in Figure 4f, with the increase in EPU content, the yield strength of thermoset materials is decreased. When the ratio of EPU to DGEBA reaches 1:1, the modified resin only undergoes high elastic–plastic deformation without yielding. Plastic deformation is mainly caused by the movement of polymer segments when the crosslinking density of EPU-modified epoxy resin is relatively low. Therefore, the originally frozen segments can begin to move under low stress. Therefore, the yield strength of modified epoxy resin gradually decreases with the increase in EPU content. Young’s modulus reflects the material’s resistance to deformation. Compared to DGEBA containing rigid benzene rings, flexible polyether chains are more prone to deformation under external forces. Therefore, both EPU and ITPU will reduce the Young’s modulus of the material, which is similar to previous research [31,37].

### 3.4. Phase Morphology

The tensile fracture morphology of samples was observed by SEM (Figure 6) to investigate the fracture mechanism. For the neat EP, a pretty smooth fracture surface with a few river-like stripes, which are typical brittle fracture characteristics, is observed [38,39]. Generally, a coarse fracture surface and severe plastic deformations are presented after adding epoxy tougheners [40,41]. However, the roughness of the surface is decreased, and a fairly flat surface is observed after the addition of EPU (Figure 6b,c), which contradicts the universal phenomenon. When an external force is applied to the EPU-modified epoxy resin, the entangled flexible polyether long chain is deformed under force (Figure 5a). Therefore, the binding of the epoxy matrix becomes small and the whole epoxy chains are easily deformed. As a result, the internal defects caused by the external force are reduced and the stress stripes of the fracture surface caused by the internal defects are diminished. Consequently, the introduction of the flexible long chain makes the tensile fracture surface smooth [42].

### 3.5. Thermogravimetric Analysis (TGA)

The thermostability of epoxy resins with the addition of the EPU was also examined, as shown in Figure 7. The 5% weight-loss temperature was defined as the initial decomposition temperature (*T*_5%_). The specific parameters from TGA and DTG are listed in Table 2. From the TGA curves and Table 3, it can be seen that the *T*_5%_ decreases with increasing EPU content. This is because EPU has many C−C and C−O bonds, which are less stable than the aromatic structure. Consequently, epoxy resins with EPU will decompose at low temperatures. However, compared to neat epoxy, the decrease is not significant. It can be seen that the *T*_5%_ of modified epoxy resin is above 330 °C, which is much higher than the processing temperature of the polymer. Therefore, the modified epoxy is stable during processing.

The peak of the DTG curve corresponds to the maximum degradation rate temperature (*T*_m_). The *T*_m_ of all samples is similar, and the content of EPU has little effect on *T*_m_. The relatively poor stability of the polyether chain in EPU results in its decomposition at a low temperature. However, since the decomposition temperature of the polyether chain and the aromatic structure in DGEBA are not significantly different, the decomposition of both EPU and DGEBA co-occurs with increasing temperature, resulting in no significant change in *T*_m_ [43]. Additionally, EPU contains more hydrogen contents, which leads to a reduction in char residue at 800 °C for the samples containing EPU [44]. However, the char residue does not decrease with an increasing amount of modifier. Therefore, the TGA results suggest excellent thermostability of the modified samples.

From Table 3, the decomposition temperature range (DTR) exhibits an increase trend as the EPU content rises, but the maximum decomposition rate (*D*_m_) exhibits a decline trend. The polyether chain in EPU has more alkyl side chains than DGEBA and D230, and these side chains begin to break down at low temperatures. The *D*_m_ relates to the decomposition of the epoxy network, which starts to break down as the temperature rises. As a result, there is a declining trend of the *D*_m_.

## 4. Conclusions

In this study, a solventless epoxy-terminated polyurethane (EPU) was synthesized and the preparation of EPU-modified epoxy resins was also environmentally friendly. It was found that the epoxy resins can be modified by any amount of EPU. In this study, the EPU content reached its highest content (50 wt%). For ITPU, liquid epoxy resin without hardener quickly formed a solid after the addition of 50 wt% ITPU. The comprehensive properties of EPU-modified epoxy resins were studied systematically and compared to the epoxy resins modified by ITPU with the same molecular structure as EPU. It was found that the toughness of epoxy resin significantly improved after the addition of EPU and the toughening effect of EPU was superior to that of ITPU. After the addition 10 wt% EPU, the elongation at break and toughness increased by 67.5% and 28.3%, respectively. In comparison, the elongation at break and toughness of epoxy resins after the addition of 10 wt% ITPU reduced by 15.81% and 7.29%, respectively. For thermal properties, they decreased after the addition of the EPU but remained at a high temperature. In addition, the thermal properties of EPU-modified epoxy resins were comparable with ITPU-modified epoxy resins at the same modifier content. These results demonstrate that the terminal epoxidation of polyurethane can solve the problems of conventional polyurethane-modified epoxy resins.

## Figures and Tables

**Figure 1 polymers-15-02803-f001:**
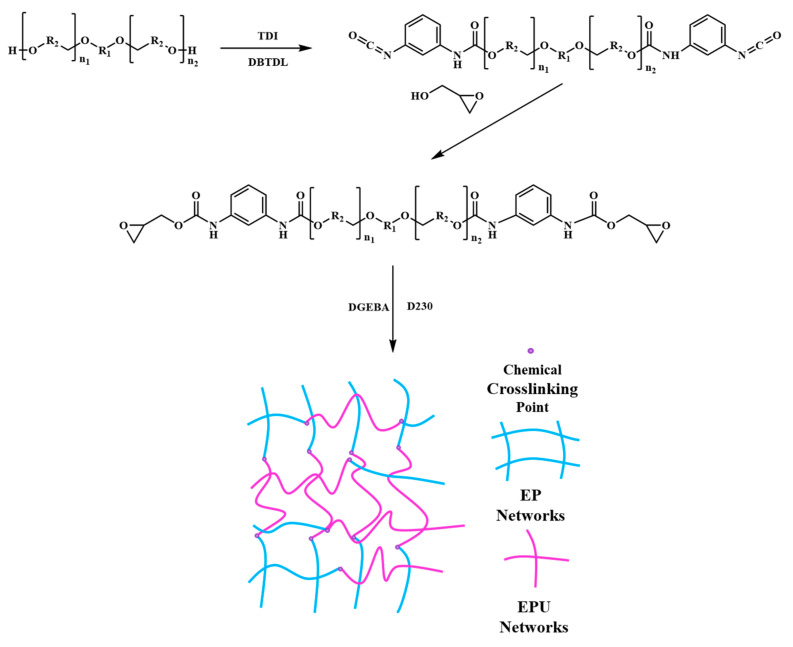
The synthesis route of epoxy−terminated polyurethane and schematic illustration of EPU-modified epoxy network.

**Figure 2 polymers-15-02803-f002:**
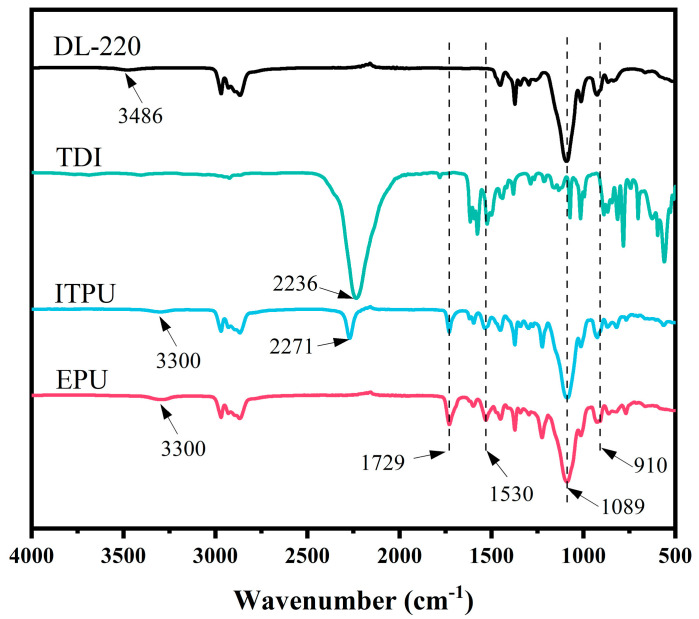
FT−IR spectra of DL−220, TDI, ITPU, and EPU.

**Figure 3 polymers-15-02803-f003:**
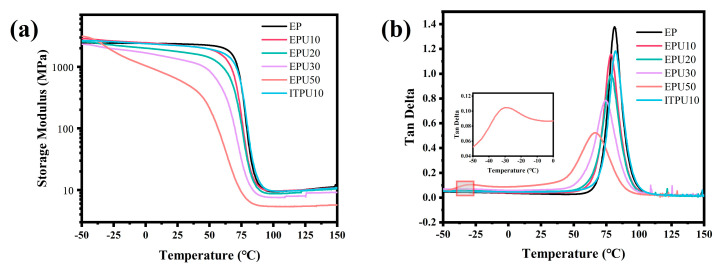
Curves from dynamic mechanical analysis: (**a**) storage modulus, (**b**) tan delta.

**Figure 4 polymers-15-02803-f004:**
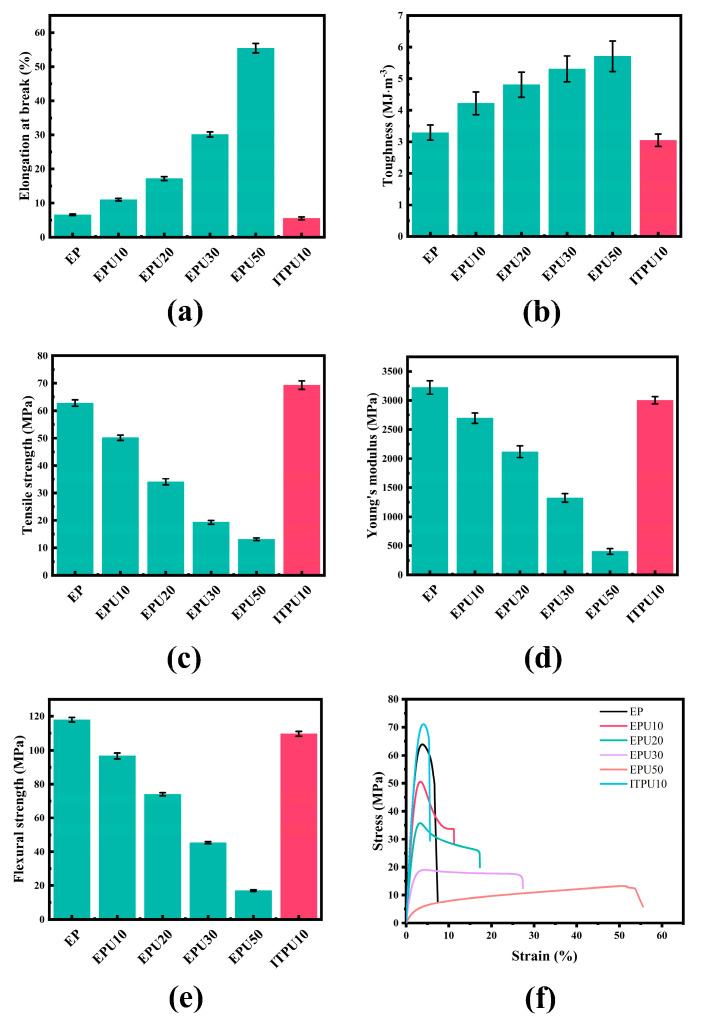
Mechanical test results: (**a**) elongation at break, (**b**) toughness, (**c**) tensile strength, (**d**) Young’s modulus, (**e**) flexural strength, and (**f**) representative stress–strain curves.

**Figure 5 polymers-15-02803-f005:**
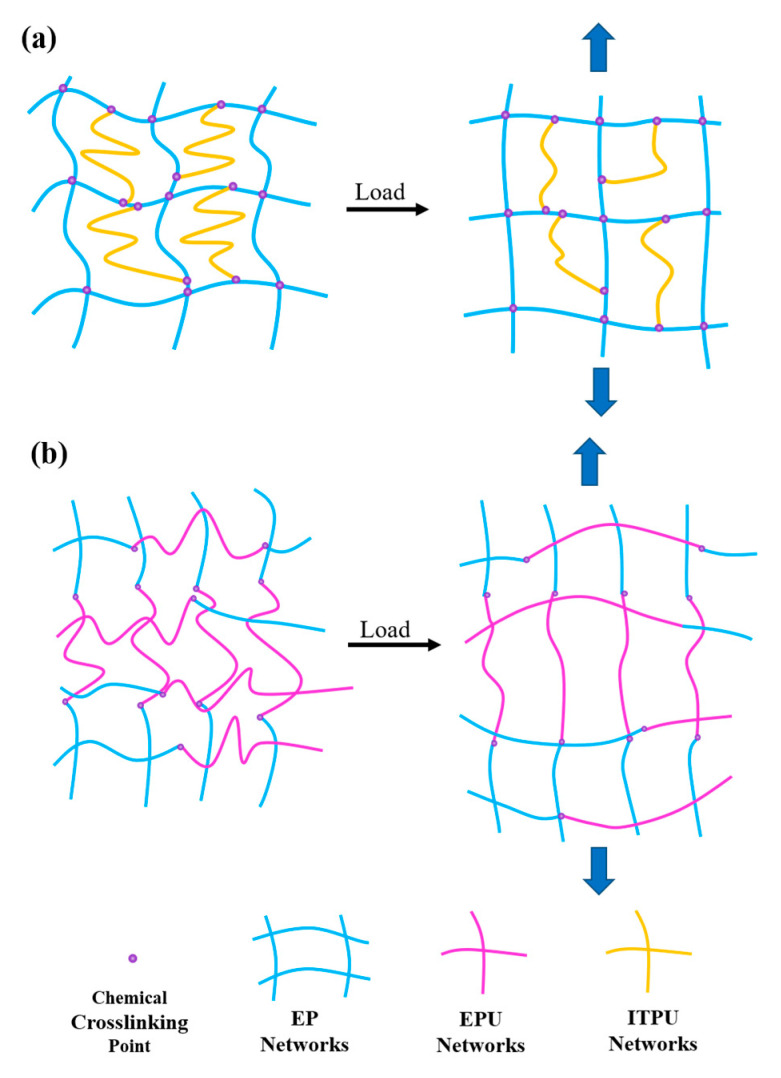
Schematic diagram of the toughening mechanism of (**a**) EPU and (**b**) ITPU.

**Figure 6 polymers-15-02803-f006:**
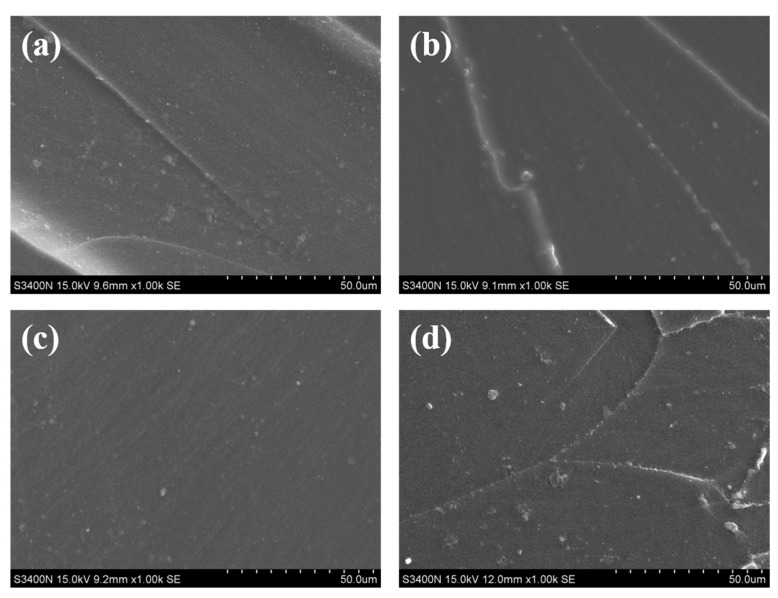
SEM images of fracture surfaces of the neat epoxy and the modified epoxy resins after tensile test: (**a**) EP, (**b**) EPU10, (**c**) EPU50, and (**d**) ITPU10.

**Figure 7 polymers-15-02803-f007:**
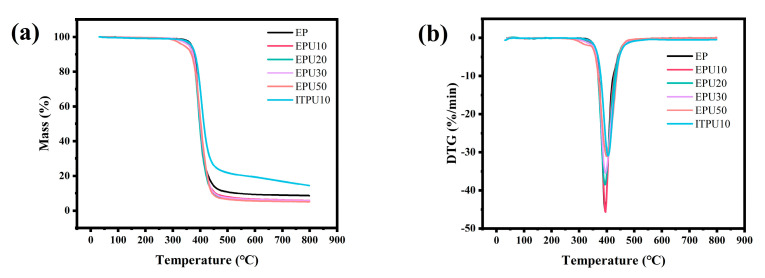
Thermogravimetric curves of curing system: (**a**) thermogravimetric curves and (**b**) derivative of thermogravimetric curves.

**Table 1 polymers-15-02803-t001:** Fabrication details of the neat epoxy resin and the modified epoxy resins.

Sample	DGEBA (g)	EPU (g)	ITPU (g)	Epoxide Value (mol (100 g^−1^)	D230 (g)
EP	100	--	--	0.51	29.33
EPU10	90	10	--	0.47	27.03
EPU20	80	20	--	0.42	24.15
EPU30	70	30	--	0.38	21.85
EPU50	50	50	--	0.29	16.68
ITPU10	90	--	10	0.45	25.88

**Table 2 polymers-15-02803-t002:** Dynamic mechanical parameters of epoxy thermosets.

Sample	*T*_g_ (°C)	*E*′ (MPa)	*ν* (10^−3^ mol·cm^−3^)	HPW (°C)	Peak Height of Tan Delta
EP	81.32	9.61	1.09	11.94	1.38
EPU10	78.73	9.73	1.02	13.57	1.16
EPU20	78.70	8.79	0.92	14.79	0.99
EPU30	74.50	7.54	0.80	18.63	0.78
EPU50	66.81	5.42	0.59	28.16	0.52
ITPU10	82.21	9.61	1.08	13.02	1.18

**Table 3 polymers-15-02803-t003:** Characteristic parameters from thermogravimetric curves.

Sample	*T*_5%_ (°C)	*T*_m_ (°C)	DTR (°C)	*D*_m_ (%/min)	Char Residue at 800 °C (%)
EP	367.07	394.2	132.5	45.29	8.55
EPU10	363.7	395.2	137.5	45.69	5.82
EPU20	360.6	393.9	145	38.58	5.58
EPU30	358.7	395.7	152.5	35.52	5.73
EPU50	339.2	400.0	170	31.09	5.11
ITPU10	368.6	405.1	133	43.73	14.42

## Data Availability

Not applicable.

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
