# Peer review of "Eco-Friendly Epoxy-Terminated Polyurethane-Modified Epoxy Resin with Efficient Enhancement in Toughness"

_polymers, 2023, doi:10.3390/polym15132803_

Round 1

Reviewer 1 Report

Comments and Suggestions for Authors

1.     The title of the publication refers to an eco-friendly epoxy-terminated polyurethane modified epoxy resin. However, the author does not explain why it is environmentally friendly.

2.     In the section «Synthesis of epoxy-terminated polyurethane», there is no evidence that the product that the authors are making is obtained.

For example, in the articles

1.     Yeganeh H., Lakouraj M. M., Jamshidi S. Synthesis and properties of biodegradable elastomeric epoxy modified polyurethanes based on poly (ε-caprolactone) and poly (ethylene glycol) //European Polymer Journal. – 2005. – Т. 41. – №. 10. – С. 2370-2379.

2.     Slobodinyuk A. et al. Synthesis and Study of Physical and Mechanical Properties of Urethane-Containing Elastomers Based on Epoxyurethane Oligomers with Controlled Crystallinity //Polymers. – 2022. – Т. 14. – №. 11. – С. 2136.

the completion of the synthesis is confirmed by analytical analyzes of the mass content of free isocyanate groups. Based on this analysis, the mass of glycidol is calculated. Next, it is necessary to carry out an analysis to determine the mass content of free epoxy groups. This analysis is necessary to calculate the required mass of hardener.

3.     The authors are studying a system that is already described in the article:

Li Y. S. et al. Polyether and polyester polyol based glycidylterminated polyurethane modified epoxy resins: Mechanical properties and morphology //Polymer international. – 1994. – Т. 35. – №. 4. – С. 371-378.

So epoxy-terminated polyurethane is synthesized by the authors of the article [Li Y. S. et al. Polyether and polyester polyol based glycidylterminated polyurethane modified epoxy resins: Mechanical properties and morphology //Polymer international. – 1994. – Т. 35. – №. 4. – С. 371-378] on the basis of polypropylene glycol with a molecular weight of 2000, 2,4 toluene diisocyanate and glycidol.

The authors of the article sent for review have the same thing. The epoxy resin used was DGEBA in both articles (or its equivalent DER-331). Moreover, the same compounds are used as the hardener: D-230 Polyoxypropyleneamine. The authors are studying mixtures of epoxy-terminated polyurethane and DGEBA (or DER-331 equivalent) cured with D-230.

The scientific novelty of the publication is not understandable and requires a more detailed explanation.

Author Response

Response to Reviewer 1 Comments

Point 1: The title of the publication refers to an eco-friendly epoxy-terminated polyurethane modified epoxy resin. However, the author does not explain why it is environmentally friendly.

Response 1: Thank you for your valuable feedback! Organic solvents are often used during the synthesis of polyurethane with low carbamate content and the fabricating of modifying epoxy resin, which contaminate soil, water, air etc. In this study, no organic solvents are required during the synthesis of EPU and the fabricating of modified epoxy. Therefore, it is environmentally friendly. In order to highlight the environmental friendliness, it is described in the last paragraph of the “Introduction part” and the “ Conclusion part”. Please see the first sentence of the last paragraph in the “Introduction part” and the first sentence of the “Conclusion part”.

Point 2: In the section «Synthesis of epoxy-terminated polyurethane», there is no evidence that the product that the authors are making is obtained.

For example, in the articles

Yeganeh, H.; Lakouraj, M.M.; Jamshidi, S. Synthesis and Properties of Biodegradable Elastomeric Epoxy Modified Polyurethanes Based on Poly(ε-Caprolactone) and Poly(Ethylene Glycol). Eur. Polym. J. 2005, 41, 2370–2379, doi:10.1016/j.eurpolymj.2005.05.004.

Slobodinyuk, A.; Strelnikov, V.; Elchisheva, N.; Kiselkov, D.; Slobodinyuk, D. Synthesis and Study of Physical and Mechanical Properties of Urethane-Containing Elastomers Based on Epoxyurethane Oligomers with Controlled Crystallinity. Polymers 2022, 14, 2136, doi:10.3390/polym14112136.

the completion of the synthesis is confirmed by analytical analyzes of the mass content of free isocyanate groups. Based on this analysis, the mass of glycidol is calculated. Next, it is necessary to carry out an analysis to determine the mass content of free epoxy groups. This analysis is necessary to calculate the required mass of hardener.

Response 2: Thank you for your valuable feedback! Actually, the amount of NCO was detected in the reaction by using the di N-butylamine technique. For detecting the reaction ITPU with glycidol, a sample of the reacting ITPU/glycidol mixture was taken at intervals of 30 min for FT-IR analysis to see whether the -NCO absorption peak is disappeared. For comparison, The FT-IR spectra of ITPU has been added, as shown in the revised Figure 2. More details for the synthesis of EPU is presented in ‘2.2 Synthesis of epoxy-terminated polyurethane (EPU)’ of the ‘Materials and Methods’ section in the revised manuscript.

Actually, the mass content of free epoxy groups was determined, as shown in Table S1 in the supporting information. As the reviewer 1 suggested, it is important for supplying this information. So, this table has now been transferred to the manuscript, please see the Table 1 in the revised manuscript.

Table 1. Fabrication details of flexible epoxy curing samples

Sample

DGEBA (g)

EPU (g)

ITPU (g)

Epoxide value (mol(100g)-1)

D230 (g)

EP

100

--

--

0.51

29.33

EPU10

90

10

--

0.47

27.03

EPU20

80

20

--

0.42

24.15

EPU30

70

30

--

0.38

21.85

EPU50

50

50

--

0.29

16.68

ITPU10

90

--

10

0.45

25.88

Point 3: The authors are studying a system that is already described in the article:

Li, Y.-S.; Li, M.-S.; Ma, C.-C.M.; Hsia, H.-C.; Chen, D.-S. Polyether and Polyester Polyol Based Glycidyl-Terminated Polyurethane Modified Epoxy Resins: Mechanical Properties and Morphology. Polym. Int. 1994, 35, 371–378,

So epoxy-terminated polyurethane is synthesized by the authors of the article [Li Y. S. et al. Polyether and polyester polyol based glycidyl‐terminated polyurethane modified epoxy resins: Mechanical properties and morphology //Polymer international. – 1994. – Т. 35. – №. 4. – С. 371-378] on the basis of polypropylene glycol with a molecular weight of 2000, 2,4 toluene diisocyanate and glycidol.

The authors of the article sent for review have the same thing. The epoxy resin used was DGEBA in both articles (or its equivalent DER-331). Moreover, the same compounds are used as the hardener: D-230 Polyoxypropyleneamine. The authors are studying mixtures of epoxy-terminated polyurethane and DGEBA (or DER-331 equivalent) cured with D-230.

The scientific novelty of the publication is not understandable and requires a more detailed explanation.

Response 3: Thanks for your comment! Polyurethane (PU) is a common polymer for constructing IPN or SIPN structures in the epoxy matrix. However, some drawbacks of PU limit its application in the modification of EP: (1) The storage condition of ITPU is harsh because of the high reactivity of isocyanate. (2) The amount of ITPU for modifying EP is limited due to the easy self-curing at high content of ITPU (Figure S1). (3) Organic solvents are often required during the polyurethane synthesis and preparing the modified epoxy resins, which is harmful to humans and environment. To address these problems, an EPU was synthesized by solventless method and no organic solvent was used during preparing the EPU modified epoxy resin (Figure 1), which is environmentally friendly. The mechanical and thermal properties of the modified epoxy resins with different EPU contents were studied systematically and compared to the modified epoxy resins with ITPU as the similar molecular structure as EPU. It was found that the toughness of the epoxy resin was significantly improved after the addition of EPU and the toughening effect of EPU outperformed that of ITPU. Meanwhile, the addition of EPU was not limited. In this study, the EPU content reached its highest content (50 wt%). For comparion, liquid epoxy resin without hardener quickly formed a solid after the addition of 50 wt % ITPU. The results shows that the terminal epoxidation of polyurethane is an efficient method to solve the problems of conventional polyurethane modified epoxy resins. For previous studies (J. Appl. Polym. Sci. 1994, 51, 1199–1206; Polym. Int. 1994, 35, 371–378), they studied the effect of EPU with various soft segment structures on the toughness of epoxy resin. The results showed that hydroxyl-terminated polyether, as a soft segment of polyurethane, can greatly increase epoxy resin's toughness. However, no research has been conducted for investigating the difference between EPU modified epoxy resins and typical ITPU modified epoxy resins to show that the terminal epoxidation of polyurethane is an efficient method for solving the problems of conventional polyurethane modified epoxy resins. As Reviewer 1 suggested, the novelty of this manuscript is highlight in the abstract and the introduction, please see lines of the Abstract and the last two paragraphs of the Introduction in the revised manuscript.

Reviewer 2 Report

The following issues must be addressed:

1.       Introduction part must be improved in order to outline what is new and innovative in this manuscript compared with other similar papers.

2.       Please explain in more details what the authors means when they claim that fewer cross-linking points make the segment less bound and movement easier.

3.       The authors claims “smooth surface” but there are no roughness values included in the manuscript.

4.       There are no prove that the internal defects caused by the external force are reduced and the stress stripes of the fracture surface caused by the internal defects are diminished. Please explain in more details.

5.       What are the grains present on the surface?

6.       Conclusion part must be significantly improved and should include the most representative results.

Author Response

Response to Reviewer 2 Comments

Point 1: Introduction part must be improved in order to outline what is new and innovative in this manuscript compared with other similar papers.

Response 1: Thanks for your comment! The Introduction part is revised to outline what is new and innovative in this manuscript compared with other similar papers, please see the last two paragraphs of the Introduction in the revised manuscript.

Point 2: Please explain in more details what the authors means when they claim that fewer cross-linking points make the segment less bound and movement easier.

Response 2: Thank you for your valuable feedback. Because the crosslinking density decreases, there are fewer chemical crosslinking points per unit volume, which means that connections between segments are weaken. As a result, when the polymer matrix is subjected to external force, there is less interaction between the segments, making it easier for the segments to move in the direction of the force.

Point 3: The authors claims “smooth surface” but there are no roughness values included in the manuscript.

Response 3: Thank you for your valuable feedback. On the fracture surface of epoxy resin, it is typically thought that the fracture surface is smooth if fewer stress stripes are visible ( ACS Appl. Nano Mater. 2021, 4, 10, 10419–10429; Industrial Crops and Products, 176, 2022, 114255 ). No researchers try to describe the morphology of fracture surfaces using roughness values because roughness value of fracture surface is hardly to be determined.

Point 4: There are no prove that the internal defects caused by the external force are reduced and the stress stripes of the fracture surface caused by the internal defects are diminished. Please explain in more details.

Response 4: Thank you for your valuable feedback. An internal defect in the material is a typical flaw where the crack on the fracture surface first appears. A stress concentration happens at the defect location in response to external force, which causes the crack to propagate. The movement of chain segments is easy after the addition of the EPU modifier because the crosslinking density is reduced, which is advantage for lowering internal defect. Internal defect characterization will be our future work.

Point 5: What are the grains present on the surface?

Response 5: Thank you for your valuable feedback. The small grains on the surface may be epoxy resin powder after fracture.

Point 6: Conclusion part must be significantly improved and should include the most representative results.

Response 6: Thanks for your comment. Conclusion part has been revised, please see the Conclusion in the revised manuscript.

Reviewer 3 Report

Dear Authors,

I attach comments in the file.

Yours sincerely,

Reviewer

Author Response

Response to Reviewer 3 Comments

Point 1: There are some minor errors edits related to unit markings, e.g. degrees Celsius

Response 1: Thanks for your comment! The errors have been corrected, please see the revised manuscript.

Point 2: The equipment used for the individual processes (mixing, drying, etc.) should be provided.

Response 2: Thanks for your comment! The equipments used for the individual processes are added, please see the ‘Materials and Methods’ part.

Point 3: Line 72. The authors mentioned about Table S1, but there is no such table in the text.

Response 3: Table S1 is the table which is put in the supporting information. Now, it has been transferred into the revised manuscript and named Table 1. In order to make all information clear, all materials from the supporting information metioned in the manuscript are marked ‘the supporting information’, please see the revised manuscript.

Table 1. Fabrication details of flexible epoxy curing samples

Sample

DGEBA (g)

EPU (g)

ITPU (g)

Epoxide value (mol(100g)-1)

D230 (g)

EP

100

--

--

0.51

29.33

EPU10

90

10

--

0.47

27.03

EPU20

80

20

--

0.42

24.15

EPU30

70

30

--

0.38

21.85

EPU50

50

50

--

0.29

16.68

ITPU10

90

--

10

0.45

25.88

Point 4: The information regarding the technological parameters and devices used mixing process should be competed.

Response 4: Thanks for your kind suggestions. The information regarding the technological parameters and devices used mixing process are added, please see the ‘Materials and Methods’ part.

Point 5: There is no information about the time and device for degassing the mixture.

Response 5: Thanks for your kind reminding! The information about the time and device for degassing the mixture are added, please see the ‘Materials and Methods’ part.

Point 6: In my opinion, the authors should include a table with the list and markings of the analyzed compositions and perhaps the composition and amount of individual components of the composition. It will be more readable.

Response 6: Thanks for your kind suggestion! The question is the same as point 3. Please see the answers for the point 3.

Point 7: It is necessary to complete the complete data of the devices (names, manufacturer, city, country).

Response 7: Thanks for your kind suggestion. The information for device are added, please see the ‘Materials and Methods’ part.

Point 8: There are no dimensions, quantities and data regarding tensile and bending strength tests in the text.

Response 8: Thanks for your kind suggestion! The dimensions, quantities and data regarding tensile and bending strength tests are added, please see the ‘Materials and Methods’ part.

Point 9: There is no information in the text on the determination of Young's modulus.

Response 9: Young's modulus is the slope of straight line at the beginning of the stress-strain curve, which can be obtained from the software of the microcomputer controlled electronic universal testing machine. In order to make it clear, some informations for obtaining Young's modulus are added to the main text, please see the last paragraph in the ‘Materials and Methods’ section.

Point 10: “tan delta” should be explained in the text, its interpretation.

Response 10: Thanks for your kind reminding. Tan delta is defined as the ratio of the loss modulus to the storage modulus. Its interpretation has been added in the revised manuscript, please see 3.2 subsection in the revised manuscript.

Point 11: There is no information on how many measurements were carried out and whether the results in Fig. 3 are the average of the number of measurements.

Response 11: Thank you for your valuable feedback. The heating rate and frequency of the instrument can affect the results of DMA. In this study, these factors are kept at the same. Therefore, each sample is tested once because the test is costly and time cosuming. And it is acceptable for testing once in basic research and industrial application.

Point 12: Line 130. Is there a mistake in table numbering? The authors refer to Table 2 while Table 1 is below.

Response 12: Thank you for your valuable feedback. It is a mistake in the manuscript. It should be Table 1.

Point 13: There is no information whether the data presented in Table 1 is the average, and the number of measurements

Response 13: This question is similar to Point 11, please see the answers of point 11.

Point 14: The authors mentioned about Table S2, but there is no such table in the text.

Response 14: Table S2 is the table in the supporting information. In order to make all information clear, all materials from the supporting information metioned in the manuscript are marked ‘the supporting information’, please see the supporting information of the revised manuscript.

Table S2 Mechanical properties of cured epoxy samples

Sample

Elongation at break (%)

Toughness (MJ·m-3)

Tensile strength (MPa)

Young’s modulus (MPa)

Flexural strength (MPa)

EP

6.58±0.23

3.29±0.24

62.75±1.16

3222±116

117.9±1.3

EPU10

11.02±0.37

4.22±0.36

50.11±1.00

2695±90

96.7±1.8

EPU20

17.22±0.55

4.81±0.40

34.06±1.14

2119±100

74.1±0.9

EPU30

30.16±0.77

5.31±0.41

19.32±0.73

1325±77

45.3±0.6

EPU50

55.42±1.36

5.71±0.48

13.16±0.49

403±48

17.0±0.5

ITPU10

5.54±0.42

3.05±0.20

69.31±1.55

3004±61

109.6±1.5

Point 15: There is no information on how many measurements were carried out.

Response 15: In the last sentence of 2.4 Characterization-“At least five replicates were tested for each group of samples and average values were reported”, the information on how many measurements were carried out is provided.

Point 16: If the authors included a list of all tested epoxide compounds in the methodological part, it would be easier to understand the interpretation of the results.

Response 16: Actually, the table (Table S1) containing these information is existed, which was put in the supporting information. Now, the table is transferred from the supporting information into the main text, please see Table 1 in the revised manuscript.

Point 17: Figure 4 point f and the text. “Representative stress-strain curves” were presented for which strength test?

Response 17: They are representative stress-strain curves from tensile tests. More information is supplied into the caption of the Figure 4, please see Figure 4 in the revised manuscript.

Point 18: Figure 5b should be in section 3.4, not in section 3.3.

Response 18: Thanks very much for reviewer 3’s suggestion! Figure 5 is useful for interpreting the change of mechanical properties. Figure 5 should be in section 3.3, in order to make the manuscript logical. This section is rewritten, please see the 3.3 section in the revised manuscript.

Reviewer 4 Report

The manuscript under the title: “Eco-friendly epoxy-terminated polyurethane modified epoxy resin with efficient enhancement in toughness” is in line with «Polymers» journal. This topic is relevant and will be of interest to the readers of the journal. It based on original research. This research has scientific novelty and practical significance. The article has a typical organization for research articles.
Before the publication it requires significant improvements, especially:

  1. The "Introduction" section: it has been proven that the effect of various modifying additives and fillers on the physicochemical and mechanical properties of epoxy polymer composites is determined by many factors: ……. I think the related references should be cited corresponding to each aspect, e.g. (but not limited to these), which will undoubtedly improve the "Introduction" section:
  • Polymer-Plastics Technology and Materials, 59, 874–883, https://doi.org/10.1080/25740881.2019.1698615
  • Polymers 202315(10), 2261; https://doi.org/10.3390/polym15102261
  • J. Compos. Sci. 20237(5), 178; https://doi.org/10.3390/jcs7050178
  • J Mater Sci 57, 9493–9507 (2022). https://doi.org/10.1007/s10853-022-07174-z
  • Appl. Polym. Sci. 2019, 136, 47410, https://doi.org/10.1002/app.47410

  1. Chinese GB standards need to be replaced by the corresponding international ISO standards.
  2. Why the heating rate (for TGA and DMA) in section 2.4. in K / min, and on the graphs the temperature is in 0C. I recommend that you present everything to 0C.
  3. Why is the amount of injected EPU limited to 50%? Why didn't you try adding more than 50%?
  4. How does the introduction of EPU affect the viscosity of the epoxy composition?
  5. Figure 7(b). as you rightly noted, Tm does not change, but it can be seen that with an increase in the EPU content in the epoxy composition, the rate of weight loss (% / min) decreases. How can you explain it?
  6. The "Conclusion" section needs to be expanded, in addition, it is necessary to indicate the scientific novelty and practical significance of the results obtained.
  7. The references contains 9 references to own works. I recommend citing your own work only where it is really necessary. I recommend reducing self-citation to 3-5 references.

Author Response

Response to Reviewer 4 Comments

Point 1: The "Introduction" section: it has been proven that the effect of various modifying additives and fillers on the physicochemical and mechanical properties of epoxy polymer composites is determined by many factors: ……. I think the related references should be cited corresponding to each aspect, e.g. (but not limited to these), which will undoubtedly improve the "Introduction" section:

  • Polymer-Plastics Technology and Materials, 59, 874–883, https://doi.org/10.1080/25740881.2019.1698615
  • Polymers 2023, 15(10), 2261; https://doi.org/10.3390/polym15102261
  • J. Compos. Sci. 2023, 7(5), 178; https://doi.org/10.3390/jcs7050178
  • J Mater Sci 57, 9493–9507 (2022). https://doi.org/10.1007/s10853-022-07174-z
  • Appl. Polym. Sci. 2019, 136, 47410, https://doi.org/10.1002/app.47410

Response 1: Thanks for your comment! In the revised manuscript, the related references has been cited.

Point 2: Chinese GB standards need to be replaced by the corresponding international ISO standards.

Response 2: Thanks for your kind suggestion. Comparing the Chinese GB standards with international ISO standards, no difference is found. As the Reviewer 4 suggested, the Chinese GB standards are replaced by the corresponding international ISO standards in the revised manuscript, please see the last paragraph in the ‘Materials and Methods section’ of the revised manuscript.

Point 3: Why the heating rate (for TGA and DMA) in section 2.4. in K/min, and on the graphs the temperature is in . I recommend that you present everything to .

Response 3: Thank you for your valuable feedback. All temperature units are changed into degrees celsius, please see the first paragraph in the subsection‘2.4 Characterization’ of the revised manuscript.

Point 4: Why is the amount of injected EPU limited to 50%? Why didn't you try adding more than 50%?

Response 4: Thank you for your valuable feedback. In this paper, EPU is used to modify epoxy resin. Therefore, the highest EPU content is 50%. When the EPU content exceeds 50%, DGEBA is employed to modify the EPU instead, which is contrary to original goals. Consequently, we set 50% as the maximum EPU content. We explain it in the revised manuscript, please see 2.3 subsection in the revised manuscript.

Point 5: How does the introduction of EPU affect the viscosity of the epoxy composition?

Response 5: Due to the high viscosity of EPU, the modified epoxy resin becomes viscous after the addition of EPU. Because this is not the main work in this paper, we didn’t conduct rheological test. The rheological investigation of EPU modified epoxy resins is ongoing, which will be included in our future work.

Point 6: Figure 7(b). as you rightly noted, Tm does not change, but it can be seen that with an increase in the EPU content in the epoxy composition, the rate of weight loss (% / min) decreases. How can you explain it?

Response 6: Thank you for your valuable feedback. Its maximum decomposition rate slows down since it has already partially broken down at low temperatures. As shown in Table R1, the decomposition temperature range exhibits an increase trend as EPU content rises, but the maximum decomposition rate exhibits a decline trend. The polyether chain in EPU has many alkyl side chains.These side chains begin to break down at low temperatures. The maximum decomposition rate relates to the decomposition of the epoxy network, which starts to break down as the temperature rises. As a result, there is a decline trend in the maximum decomposition rate. This explaination is added to the main text and some parameters (DTR and Dm) are added to the Table, please see the last paragraph of the ‘Results and Discussion’ section and Table 3.

Table R1 Characteristic parameters from thermogravimetric curves

Sample

T5% (℃)

Tm (℃)

DTR (℃)

Dm (%/min)

Char residue at 800℃ (%)

EP

367.07

394.2

132.5

45.29

8.55

EPU10

363.7

395.2

137.5

45.69

5.82

EPU20

360.6

393.9

145

38.58

5.58

EPU30

358.7

395.7

152.5

35.52

5.73

EPU50

339.2

400.0

170

31.09

5.11

ITPU10

368.6

405.1

133

43.73

14.42

Point 7: The "Conclusion" section needs to be expanded, in addition, it is necessary to indicate the scientific novelty and practical significance of the results obtained.

Response 7: Thanks for your comment. Conclusion part has been revised, please see the Conclusion in the revised manuscript.

Point 8: The references contains 9 references to own works. I recommend citing your own work only where it is really necessary. I recommend reducing self-citation to 3-5 references.

Response 8: Thanks for your comment. We have reduced self-citation to 4 references, please see the references in the revised manuscript.

Reviewer 5 Report

The manuscript “Eco-friendly epoxy-terminated polyurethane modified epoxy resin with efficient enhancement in toughness” may be published in Polymers after revision.

Remarks

1. Introduction

The introduction needs to be expanded. Note the importance of modifying epoxy resins to give them improved properties (e.g. epoxy resins with increased flame retardance https://doi.org/10.3390/polym14173592, https://doi.org/10.3390/polym14245334).

2. Materials and methods

Specify how the impact strength of EPU was determined? ASTM standard method according to Izod? Indicate the characteristics of the machine with which the impact resistance was determined. How much polyesteramine, DGEBA and EPU were added to make the cured resin? Include downloads in the main body of the article, not in the Supplementary materials.

3. Results

What is the reason for the lower Flexural Strength and Tensile Strength of the obtained EPUs compared to EPs? (Figure 4).

4. Conclusions

Where do you plan to use the received EPU? What research do the authors plan to conduct in the future?

Minor English editing required

Author Response

Response to Reviewer 5 Comments

Point 1: Introduction

The introduction needs to be expanded. Note the importance of modifying epoxy resins to give them improved properties (e.g. epoxy resins with increased flame retardance https://doi.org/10.3390/polym14173592, https://doi.org/10.3390/polym14245334).

Response 1: Thanks for your comment. I’m sorry to tell that we cannot discuss the flame retardance because it is not topic of our paper. But the literatures mentioned above has been cited, please see the references in the revised manuscript.

Point 2: Materials and methods

Specify how the impact strength of EPU was determined? ASTM standard method according to Izod? Indicate the characteristics of the machine with which the impact resistance was determined. How much polyesteramine, DGEBA and EPU were added to make the cured resin? Include downloads in the main body of the article, not in the Supplementary materials.

Response 2: Thank you for your valuable feedback. Impact experiments were not performed in this study due to the experimental conditions. The toughness data in this research is the integral area of the stress-strain line integral of the tensile test, which can be seen in the last paragraph of the revised manuscript.

The table containing fabrication details has been transferred from the supporting information  to the manuscript, please see the Table 1 in the revised manuscript.

Point 3: Results

What is the reason for the lower Flexural Strength and Tensile Strength of the obtained EPUs compared to EPs? (Figure 4).

Response 3: Thank you for your valuable feedback. The crosslinking density of epoxy resin diminishes with the addition of EPU. Lower crosslinking density lowers tensile strength and bending strength.

Point 4: Conclusions

Where do you plan to use the received EPU? What research do the authors plan to conduct in the future?

Response 4: We plan to use EPU to toughen epoxy resins for appling in high-toughness-required fields such as aircraft structures, space structures, cryogenic engineering and so on. The molecular structure of the toughener will be changed in the future to preserve the current level of toughness while reducing the negative effect on other properties including strength, modulus, glass transition temperature, thermal decomposition temperature, etc.

Round 2

Reviewer 2 Report

The manuscript can be published in present form.

Reviewer 3 Report

Dear Authors,

I hereby accept the corrections  in resubmitted manuscript.

Yours sincerely,

Reviewer

Reviewer 4 Report

The authors considered most of the comments or adequately responded to the remarks contained in the review; therefore, the work may be approved for publication.

Reviewer 5 Report

The authors took into account all comments. I recommend the manuscript for publication.

English needs some editing